# Adult Dyslexia Traits as Predictors of Hot/Cool Executive Function and Prospective Memory Abilities

**DOI:** 10.3390/brainsci15101065

**Published:** 2025-09-29

**Authors:** Christina Protopapa, Rachael L. Elward, James H. Smith-Spark

**Affiliations:** School of Allied Health and Life Sciences, College of Health and Life Sciences, London South Bank University, 103 Borough Road, London SE1 0AA, UK; protopc2@lsbu.ac.uk (C.P.); elwardr@lsbu.ac.uk (R.L.E.)

**Keywords:** developmental dyslexia, executive functioning, prospective memory, adult cognition

## Abstract

**Background/Objectives**: Executive functioning (EF) and prospective memory (PM) difficulties have been documented in adults with developmental dyslexia. However, research has tended to focus on university students with a formal diagnosis, overlooking adults who may experience symptoms of dyslexia without ever being formally diagnosed. The present online study aimed to investigate the effects of dyslexia traits on EF and PM in a larger, community-based sample by prioritising the use of self-report measures that include and differentiate between underexplored aspects of EF and PM in the dyslexia literature. **Methods**: One hundred and eighty adult volunteers of a wide range of ages were included in the final analysis. Participants completed four self-report questionnaires with good pedigrees assessing dyslexia traits and ADHD symptomatology, as well as everyday experiences of five distinct EFs, PM and PM strategies. **Results**: Hierarchical regression analyses revealed that, after controlling for age and ADHD symptomatology, more self-reported dyslexia traits were associated with more frequent EF problems overall and lower confidence in PM Abilities. Elevated dyslexia traits were significantly associated with self-reports of more difficulties with time-management, organisation and self-restraint, but not with more difficulties with motivation and regulation of emotion. No significant associations were found between dyslexia traits and the self-reported use of PM-enhancing strategies. **Conclusions**: The findings are considered in the light of potential metacognition deficits and theoretical and practical implications are discussed.

## 1. Introduction

Developmental dyslexia is a specific learning difficulty (SpLD) associated with problems with reading and/or spelling [1] and frequently co-occurs with other SpLDs, such as Attention Deficit Hyperactivity Disorder (ADHD; [2]). The dominant theories of dyslexia have focused on explaining the phonological deficits associated with the condition (e.g., [3,4]) and, as such, numerous reading and spelling interventions have been developed to address the literacy difficulties experienced by children and young adults with dyslexia (for reviews, see [5,6]). However, traditional theories of dyslexia fail to take into account the sustained negative impact that dyslexia can have on various everyday cognitive processes, such as executive functioning (EF) and prospective memory (PM), in both children (e.g., [7,8,9,10,11]) and adults (e.g., [8,12,13,14,15]).

Executive functioning and PM are higher-order cognitive abilities; they are complex mental processes that extend beyond the basic cognitive functions of memory, attention and perception [16]. They are considered “higher-order”, because they involve the integration of multiple cognitive processes in order to adapt to more challenging situations in everyday life and achieve goal-directed behaviour (e.g., [16,17]). As such, they are involved in advanced thinking, understanding, reasoning and problem-solving, which are crucial for navigating complex tasks and making decisions (e.g., [18,19]). It is generally agreed that there are three core executive functions (EFs): working memory, inhibitory control and cognitive flexibility, from which higher-order EFs can be built, such as planning, problem solving and reasoning [20,21]. Executive functioning enables individuals to prioritise, plan activities, switch between tasks, resist distractions and manage emotions. These processes are associated with prefrontal cortex (PFC) networks, mainly dorso- and ventrolateral PFC, although multiple other brain areas may be co-activated depending on the particular EF being utilised (e.g., [20,22,23]). Prospective memory is called upon to remember to carry out intentions at the appropriate time in the future [24], such as attending appointments, taking medication and meeting deadlines. The rostral PFC plays a key role during intention formation, maintenance and execution, although other brain regions, such as the parietal lobe and anterior cingulate, may also be activated, at least during laboratory paradigms of PM [25]. Although EFs and PM are separate constructs, they draw on similar cognitive mechanisms and the demands they make on cognition can overlap. For instance, planning an intention (requiring EF) and remembering to execute it at the appropriate time (requiring PM) calls for both processes to work in tandem [26]. For individuals with developmental dyslexia, the higher-order cognitive abilities, of EF and PM, have been found to be impaired, this is the case for both children (e.g., [7,8,9,10,11]) and adults (e.g., [8,12,13,14,15]). While adults with dyslexia can avoid literacy-related tasks post education, they still need to operate in situations calling upon EFs and PM in their everyday lives. Such abilities are fundamental for managing daily demands in adulthood and play a vital role in the successful everyday functioning of adults, both in a personal and professional capacity [20]. However, historically, dyslexia often went unrecognised within educational settings, therefore many adults, particularly those now in middle and older adulthood, may have progressed through education without their problems being noticed or addressed [27], yet they may still experience consequences in everyday cognition at present. Therefore, further investigation is warranted in these two areas of cognition in relation to dyslexia traits (not necessarily confirmed diagnosis), to develop a greater understanding of the association between traits associated with dyslexia and strengths or difficulties in the everyday use of EF and PM. This understanding may also contribute to the identification of potential indicators of dyslexia in undiagnosed adults, particularly those whose work or daily activities involve little reading or writing, possibly due to a conscious effort to avoid an area of difficulty.

### 1.1. Executive Functioning in Dyslexia

Difficulties with EF in individuals with formally diagnosed dyslexia are well-documented in the literature. Brosnan et al. [8] used a range of laboratory-based tasks to explore the effects of dyslexia on EF in both children and young adults. Compared with those without dyslexia, individuals with dyslexia across both age groups had greater difficulty in tasks where contextual distractors had to be ignored, suggesting a deficit in the EF of inhibition. Similarly, using a combination of self-report and laboratory methods, Smith-Spark et al. [12] investigated a broad range of EFs in adults with and without dyslexia. The results from the laboratory measures were consistent with Brosnan et al. [8] and indicated that adults with dyslexia performed worse on tasks requiring inhibition and also on tasks demanding updating and switching. The self-reports indicated that adults with dyslexia tended to report more frequent EF difficulties in their everyday lives, as indicated by their responses on the adult version of the Behaviour Rating Inventory of Executive Function (BRIEF-A; [28]). In particular, they reported more problems in metacognitive aspects of EF, such as experiencing problems with working memory, planning and organisation. No significant differences, however, were reported in the ability to regulate behavioural and emotional responses. These self-report data were consistent with Smith-Spark et al.’s [29] finding that, compared with controls, adults with dyslexia self-reported more frequent everyday cognitive failures on the Cognitive Failures Questionnaire (CFQ; [30]), particularly on CFQ items closely linked to EF failures in terms of aspects of attentional control, but not on items which relate to the control of emotion. Evidence from more ecologically valid paradigms further supports the argument that individuals with dyslexia may experience difficulties with EF. Using the Jansari Assessment of Executive Function (JEF; [31]), a non-immersive virtual reality office environment, Smith-Spark et al. [32] found that, compared with controls, participants with dyslexia exhibited more difficulty in applying the EFs of planning and selective-thinking to work-related tasks and performed at a lower level overall.

Jointly, these findings are indicative of a potential distinction between the ‘type’ of EF that may be impaired in dyslexia. Zelazo and Müller [33] proposed that EFs exist on a spectrum, ranging from “cool” to “hot”. Cool EFs refer to skills used in decontextualized tasks with no motivational significance (e.g., time-management), whereas hot EFs are involved in the processing of information related to emotion, reward and motivation. However, classifying individual EFs as being purely hot or cold is complex, as some processes (i.e., inhibition) encompass multiple modalities [20,34] and, therefore, create an overlap in demands on both hot and cool EFs (e.g., [35,36]). For instance, inhibition involves both regulating actions and the underlying emotions driving those actions ([37], see [38], for a discussion of the interplay between EF and emotion regulation). This co-involvement of cool and hot EF processes suggests that such processes can be better understood as existing along a continuum rather than as distinct categories [33] and, moreover, any classification may also vary depending on the context (e.g., the motivational significance of the problem) and task demands (e.g., whether the problem itself is cool or hot, [36,38]).

Research discussed thus far suggests that individuals with dyslexia may have difficulties with cool EFs, but their hot EFs may remain unaffected. While socioemotional difficulties in dyslexia are well documented (e.g., [39,40,41]), research has tended to focus on emotions from a social–psychological perspective (e.g., self-esteem [39,40]), rather than addressing emotional regulation from a more cognitive direction. To explore this argument further, the present study utilised the Barkley Deficits in Executive Functioning Scale (BDEFS; [42]), a well-established, clinical self-report measure of EF deficits in everyday life. The scale measures EF deficits across five everyday domains that accurately represent the idea that EFs run on said spectrum and can be classified into distinct categories depending on context. Two factors, namely Self-Management to Time and Self-Organisation/Problem Solving, represent what, in the authors’ opinion, would be considered cool EFs and three factors, namely Self-Restraint/Inhibition, Self-Motivation and Self-Regulation of Emotion, represent what would be considered hot EFs (N.B. this classification is based on the current authors’ judgement of the degree of emotional and motivational significance of the questions, not on published guidelines provided by the author of the scale). Collectively, these components provide an overall EF profile score. Barkley [42] argued for the ecological validity of the BDEFS, in comparison with laboratory EF tests, highlighting that the context in which those latter tests are used is not an accurate representation of everyday life activities. So, although EF tests can be effective in evaluating the most basic, immediate and cognitive aspects of EF, they are poor at measuring the higher-level adaptive, tactical and strategic dimensions of EF as manifested in everyday behaviour and social interactions—both of which unfold over more extended timeframes. This view aligns with Stanovich’s [43] distinction between the reflective and algorithmic levels of cognition. The reflective level relates to reviewing one’s beliefs in order to act appropriately and achieve goals, whereas the algorithmic level involves cognitive mechanisms that interpret environmental cues to apply correct beliefs and guide actions. Given that self-reports are grounded in an individual’s lived experiences (i.e., at the reflective level), they could potentially be capturing more complex, context-specific aspects of EF, thereby providing a more comprehensive understanding of the everyday use of EFs.

### 1.2. Prospective Memory in Dyslexia

Prospective memory difficulties in individuals with dyslexia are also highlighted in the literature. Khan [9] investigated PM in children with dyslexia, administering the Prospective and Retrospective Memory Questionnaire (PRMQ; [44]). The PRMQ is well-established and accurately measures PM Abilities by considering both the retrospective and prospective elements involved in PM remembering. Einstein and McDaniel [45] proposed that, although a PM component is necessary to remember to execute planned intentions at an appropriate time, a retrospective memory component is also necessary in order to retain the intention, until it is time to be executed. Therefore, successful prospective remembering relies on both prospective and retrospective elements working in tandem. Khan [9] found that children with dyslexia significantly reported more PM failures than children without dyslexia. Smith-Spark et al. [13] administered the PRMQ to adults with dyslexia and found that difficulties with PM in dyslexia are also experienced in adulthood. To corroborate the participants’ self-reports, they also collected proxy ratings from close associates, which again indicated more frequent everyday PM failures in adults with dyslexia. Smith-Spark et al. [15] further advanced the methodological approaches used in previous research by looking at both the typical, everyday PM performance of adults with dyslexia (measured by self-reports) and their optimal performance on PM tasks under laboratory conditions. In so doing, the authors found evidence of dyslexia-related PM deficits at both the reflective level (evidence from self-reports) and algorithmic level (evidence from laboratory measures) of cognition (see [43]). Further to this, Smith-Spark et al. [15] incorporated a naturalistic task in their study, requiring participants to leave a voicemail for the researcher 24 h after the study had finished. Adults with dyslexia were significantly more likely to forget to carry out this task, compared with adults without dyslexia. Similarly, Smith-Spark et al. [3] found that under non-immersive virtual reality conditions, adults with dyslexia had more difficulty applying PM to work-related tasks compared with controls, providing additional evidence for PM problems under naturalistic conditions (see also [14]) and further corroborating the findings from the self-report and experimental measures.

As discussed, individuals with dyslexia exhibit difficulties in laboratory tasks of PM and also self-report more PM difficulties in everyday life (e.g., [9,13,15]). However, little is known about whether individuals with dyslexia make use of any intention offloading strategies to enhance prospective remembering. Intention or cognitive offloading refers to the use of external aids to reduce the cognitive load associated with processing and retaining information [46,47]. For example, using a calendar or digital reminder to remember future appointments can preserve mental resources for other ongoing activities requiring cognitive demand. It is important to consider such strategies when studying PM in an ecologically valid manner, as everyday PM performance heavily relies on these strategies and this assumption has been supported by evidence from naturalistic and self-report studies of PM (e.g., [48,49,50]).

The decision to offload an intention is guided by an individual’s awareness and understanding of their own memory strengths and limitations, also known as metamemory (e.g., [51,52]). Good metamemory skills can mitigate the risk of forgetting by enabling individuals to recognise situations where potential memory failures may occur [53] and then apply offloading strategies accordingly. Only a very small number of studies have explored metacognitive awareness and strategy use in relation to the cognitive difficulties experienced by individuals with dyslexia [15,54,55,56]). To address this gap in the literature, the present study utilised the short version of the Metacognitive Prospective Memory Inventory (MPMI-s; [57]), a self-report measure of PM Abilities that also assesses internal and external strategy use relating to PM-related intention offloading. This inventory contains items measuring both prospective forgetting (as most self-report PM measures) and prospective remembering. Moreover, it is the first self-report measure to incorporate both internal and external PM strategies in a single inventory. Internal PM strategies involve using mnemonic techniques to remember information, such as repeating information to oneself or using mental images of the to-be-remembered task, whereas external PM strategies are intention offloading strategies that rely on memory cues in the environment, such as writing lists and using alarms [50,58,59].

### 1.3. The Present Study

The present study aimed to use ecologically valid, self-report measures to expand the current knowledge on the relationship between dyslexia and EF and PM deficits in everyday life, and the strategies employed to overcome these deficits. To the best of the authors’ knowledge, the BDEFS and MPMI-s have not been used in relation to dyslexia before. Following a paradigm similar to that of Protopapa and Smith-Spark [60], who looked at dyslexia traits rather than official dyslexia diagnosis, the current online self-report study explored how self-reported dyslexia traits related to EF, PM and PM-related strategy use. Considering the frequent co-occurrence between dyslexia and ADHD [2], and the well-documented executive function difficulties associated with ADHD (e.g., [35,61,62,63]), co-occurring self-reported ADHD symptomatology was statistically controlled in this study. As discussed, research on the impact of dyslexia on various aspects of cognition in adulthood is limited and has tended to focus on the experiences of young, university students (e.g., [12,13,14,29]). To address these dual issues, the present study aimed to recruit respondents from a wider, community-based sample of various ages and educational backgrounds to document the self-reports of adults who may experience symptoms of dyslexia in the absence of a formal diagnosis. Given the aim to recruit a much wider age range than typically assessed in the dyslexia literature, and considering evidence suggesting that various EFs decline during ageing (e.g., [64,65,66,67]), participant age was also statistically controlled in the analysis.

It was firstly hypothesised that participants who self-report a higher incidence of traits related to dyslexia would also report more EF difficulties overall, even after controlling statistically for age and co-occurring ADHD symptomatology. Secondly, and more specifically, it was hypothesised that participants self-reporting more dyslexia-related traits would also report more difficulty on the Self-Management to Time and Self-Organisation/Problem Solving subscales of the BDEFS. These predictions were based on the previous literature on the role of diagnosed dyslexia in EF (e.g., [8,12]). Finally, due to a lack of definitive empirical evidence relating to the remaining variables of the BDEFS to dyslexia, with the exception of Smith-Spark et al. [12], who found no group differences in behaviour regulation on the BRIEF-A [28], it was hypothesised that there would be no association between the incidence of dyslexia-related traits and difficulty in Self-Restraint, Self-Motivation and Self-Regulation of Emotion. In sum, it was predicted that dyslexia-related traits would be associated with deficits in cool EFs, but not deficits in hot EFs.

With regard to PM, it was firstly hypothesised that, after controlling for age and ADHD symptomatology, participants who self-report higher levels of dyslexia traits would report more frequent PM difficulties, by means of low ratings in regard to their PM Abilities on the MPMI-s. Secondly, the relationship between dyslexia symptoms and PM Abilities was predicted to be negative, in line with the profile of PM difficulties in diagnosed dyslexia [13,14,15]. Thirdly, it was hypothesised that a higher incidence of self-reported dyslexia traits would be associated with a more frequent self-reported use of PM strategies, if not more effective self-reported PM functioning (see [15]).

## 2. Materials and Methods

### 2.1. Participants

The initial sample consisted of 304 adults recruited online from social media adverts placed on LinkedIn, Facebook, Reddit and Instagram (N = 186) or from the Research Participation Scheme (RPS) subject pool at the authors’ host institution (N = 118). All participants were aged between 18 and 61 years (mean = 25.70, *SD* = 7.84, N = 17 did not disclose their age). Participation was voluntary and no compensation was offered to the volunteers, with the exception of those recruited through the RPS system, who received course credit for taking part. The only requirement for participation was that participants were over the age of 18 and did not have a diagnosis of autism or any genetic or neurodevelopmental disorder that may be associated with cognitive or intellectual impairment. Individuals with specific learning differences (i.e., dyslexia, ADHD, developmental coordination disorder, dyscalculia and dysgraphia) were welcome to participate.

Following data collection and to ensure consistency in the interpretation of the questions, 85 participants were excluded due to being non-native English speakers (all questionnaires were presented in English, see [68] for a discussion on response style differences between countries). Twenty-seven participants were excluded on the basis of providing incomplete responses to the questionnaires (i.e., if one or more questions were not answered). A further participant was excluded because of straightlining (i.e., the same response was selected in most questions, challenging the validity of response). Eight participants were excluded due to reporting having a diagnosis of autism or a diagnosis of an intellectual impairment, both of which were clearly identified as exclusion criteria in the participant recruitment and briefing statements. Although participants with a diagnosis of other SpLDs (other than dyslexia and ADHD that was controlled for in the study) were initially welcome to participate, a further three participants with reported diagnoses of developmental coordination disorder or dyscalculia were excluded in order to maintain a more homogeneous sample and to reduce the potential influence of additional co-occurring symptoms on the findings.

The final sample, therefore, consisted of 180 adult volunteers (145 female, 33 male, 1 nonbinary, 1 gender-fluid) aged between 18 and 61 years (mean = 25.70, *SD* = 8.93, N = 11 did not disclose their age). All the participants were native English speakers and 22 participants reported having more than one native language. Twelve participants in the sample reported having an official diagnosis of dyslexia, while three participants considered themselves to have dyslexia without receiving official confirmation. Ten participants reported having a diagnosis of ADHD, two of whom also reported co-occurring dyslexia and one reported a self-declared dyslexia status. A summary of the other background measures obtained from the participants is presented in Table 1.

### 2.2. Materials

The Qualtrics XM platform was used to record responses.

The Adult Reading Questionnaire (ARQ; [69]) is a 15-item scale with modest to good reliability reported across its factors (all *α* > 0.58; [69]), used to assess characteristics associated with dyslexia in adult respondents. Four items ask about a dyslexia diagnosis and two items require respondents to rate the frequency with which they read and write in everyday life. The remaining nine items require respondents to rate “symptoms” typically associated with dyslexia, such as literacy skills, organisation and word-finding. Seven of those items are rated on a five-point Likert scale ranging from never to always (e.g., “Do you find it difficult to read aloud?”), whilst the rest of the items are rated on varying scales. According to the original authors’ published scoring guidelines, the minimum possible score on the ARQ is 0 and the maximum is 43.

The ADHD Self-Report Scale (ASRS; [70]) is an 18-item scale with high internal consistency (α > 0.88; [71]) used to screen for ADHD symptomatology in adults. In the current study, only Part A was used (acceptable internal consistency *α* > 0.60; [72]), which comprised six items that were identified by the authors as the strongest predictors of symptoms associated with ADHD. Four items tap into difficulties with sustained attention, PM and organisation (e.g., “How often do you have difficulty getting things in order when you have to do a task that requires organisation?”) and two items target hyperactivity (e.g., “How often do you fidget or squirm with your hands or feet when you have to sit down for a long time?”). The six items are rated on a five-point Likert scale ranging from “never” to “very often”. The minimum score for this measure is zero and the maximum score is six, in line with the scoring instructions for the frequency of occurrence of symptoms.

The Barkley Deficits in Executive Functioning Scale (BDEFS; [42]) is an 89-item self-report scale with high internal consistency (*α* > 0.91; [73]), used to assess EF deficits in adults across five distinct EF dimensions. Respondents select the response that best describes their behaviour during the past six months, rating the frequency with which they experience each given problem on a four-point Likert scale ranging from “never or rarely” to “very often”. The first factor, “Self-Management to Time”, is concerned with an individual’s sense of time in terms of management and planning, as well as preparing for goal-directed behaviour. It consists of 21 items (e.g., “Procrastinate or put off doing things until the last minute”) and produces a minimum score of 21 and a maximum score of 84. The second factor, “Self-Organisation/Problem Solving”, relates to action, thought and writing organisation, as well as solution invention and rapid thinking. It consists of 24 items (e.g., “I do not seem to anticipate the future as much or as well as others”) and yields a minimum score of 24 and a maximum score of 96. The third factor, “Self-Restraint/Inhibition”, is concerned with impulsive decision making and behaviours, and poor inhibition of emotions and reactions. It consists of 19 items (e.g., “Find it difficult to tolerate waiting; impatient”) and yields a minimum score of 19 and a maximum score of 76. The fourth factor, “Self-Motivation”, relates to the effort and time put into assigned work and consists of 12 items (e.g., “Likely to take short cuts in my work and not do all that I am supposed to do”) that can yield a minimum score of 12 and a maximum score of 48. The fifth and final factor, “Self-Regulation of Emotion”, relates to the inhibition of strong emotional responses and self-regulatory actions relating to emotion. It consists of 13 items (e.g., “Quick to get angry or become upset”) and generates a minimum score of 13 and a maximum score of 52. The BDEFS also provides an overall EF summary score, with the minimum score being 89 and the maximum score being 356.

The short version of the Metacognitive Prospective Memory Inventory (MPMI-s; [57]) is a 22-item measure of self-reported prospective memory (PM) abilities and strategy use in relation to improving the efficacy of PM. It comprises three scales: PM Abilities, Internal PM Strategies and External PM Strategies. Its authors reported good reliability for all scales (*α* ≥ 0.70). Respondents are required to rate the frequency with which they experience each item on a five-point Likert scale ranging from “rarely” to “often”. The PM Abilities scale consists of eight items; four relate to prospective remembering (e.g., “I remember my appointments which are coming up in a few days without writing them down”) and four relate to prospective forgetting (e.g., “I forget to cancel contract on time, like trial subscriptions for newspapers”). Prospective forgetting items are reverse coded such that higher scores reflect better PM Abilities. The use of reverse-keyed items minimises potential wording effects and acquiescence bias, which could influence participant responses and compromise their validity [74]. The minimum score for the PM Abilities component is 0 and the maximum is 40. The strategy scales consist of seven items each, relating to Internal PM Strategies (e.g., “Even when I’m busy doing other things, I deliberately try to keep unfinished tasks in mind so that I do not forget them”) or External PM Strategies (e.g., “I write myself a to-do list to remind me of things that I still need to accomplish”). The minimum score for each of the strategies components is 0 and the maximum is 35, with higher scores suggesting a higher propensity to use internal and/or external PM strategies.

### 2.3. Data Analysis

Separate hierarchical multiple regression analyses were carried out following a similar method to that of Protopapa and Smith-Spark [60]. The three predictor variables were the participant’s self-declared age and their scores on the ARQ [69] and the ASRS [70], with the latter two being the screening tools for dyslexia and ADHD symptomatology, respectively. Higher scores on the predictor variables indicated a higher incidence of either dyslexia traits or ADHD symptomatology.

For all analyses, using the Enter method, participant age was entered as a predictor in Block 1, ASRS scores were entered in Block 2 and ARQ scores were entered in Block 3. The criterion variables (which were analysed in separate hierarchical regressions) were the measure of EF ability provided by the BDEFS and its sub-components (i.e., Self-Management to Time, Self-Organisation/Problem Solving, Self-Restraint/Inhibition, Self-Motivation and Self-Regulation of Emotion), the measure of PM ability was provided by the MPMI-s, and the measure of Internal or External PM strategy use was provided by the MPMI-s. Higher scores on the BDEFS and its subcomponents indicated an increased frequency of self-reported EF problems. Lower scores on the PM Abilities component of the MPMI-s indicated an increased frequency of self-reported PM deficits. Higher scores on the strategy components of the MPMI-s indicated a more frequent use of internal or external PM strategies.

In sum, the analyses were used to determine whether dyslexia traits were a significant predictor of EF and/or PM deficits, as well as the use of internal and/or external PM strategies, after accounting for age and ADHD symptomatology.

### 2.4. Procedure

The study received ethical approval from the School of Applied Sciences research ethics committee at the authors’ host institution. The participants were first presented with the information brief that included details about the procedure and purpose of the online survey. They were required to provide informed consent prior to the presentation of the questionnaires. Following consent, the participants provided demographic information and completed the four self-report questionnaires in the following order: the ARQ, the ASRS, the BDEFS and the MPMI-s. Upon completion of the questionnaires, the participants were presented with a written debrief that included contact details of the researcher and dyslexia support services in the case of concerns.

## 3. Results

Descriptive statistics for the measures used, including mean, standard deviation and minimum and maximum participant scores are summarised in Table 2. Reliability coefficients for the measures were also calculated and are presented in Table 2. All Cronbach’s α coefficient values ranged from Acceptable to Excellent, at ≥0.74 [75].

### 3.1. Correlational Analysis

Pearson’s correlations were carried out to determine the bivariate relationships between the measures in the study. Table 3 shows the correlations between participant age, ARQ score, ASRS score, BDEFS total and subtotal scores and MPMI-s subtotal scores.

There were significant (all *p*-values ≤ 0.020), but weak negative correlations between age and both the ARQ and Overall BDEFS scores, as well as between age and three sub-components of the BDEFS (namely Self-Organisation/Problem Solving, Self-Motivation and Self-Regulation of Emotion) and a significant, weak positive correlation between age and the External PM Strategies component of the MPMI-s, *p* = 0.004. There were no other significant correlations involving age (all *p*-values ≥ 0.070).

There were significant weak to moderate correlations between the ARQ and all measures (all *p*-values < 0.001), apart from the Internal and External strategy components of the MPMI-s (*p*-values ≥ 0.252). Higher dyslexia-related traits tended to be associated with higher ADHD-reported symptoms and more frequent EF problems across all five domains of the BDEFS. Higher levels of dyslexia traits also tended to be associated with lower self-reported PM Abilities. Similarly, the ASRS was significantly and moderately to strongly associated with all measures (all *p*-values < 0.001), apart from the Internal and External strategy components of the MPMI-s (*p*-values ≥ 0.098). Higher ADHD-related symptoms tended to be associated with more EF difficulties reported on all subscales of the BDEFS, as well as lower PM Abilities reported on the MPMI-s.

There were moderate to strong highly significant correlations between BDEFS Overall scores and scores on the subcomponents (all *p*-values < 0.001). There were significant weak to moderate negative correlations between the BDEFS and its subcomponents and the Abilities component of the MPMI-s (all *p*-values < 0.001), indicating that lower self-reported PM Abilities tended to be associated with more frequent EF difficulties. There were no significant associations between the BDEFS and its subcomponents and the Internal and External PM strategy components of the MPMI-s (all *p*-values ≥ 0.278), apart from a very weak negative correlation between Self-Management to Time and External PM strategies, *p* = 0.047. More frequent self-reported difficulties in time management tended to be associated with the use of less external PM strategies.

There were highly significant, weak positive correlations between the PM Abilities component of the MPMI-s and the Internal and External PM strategies components (all *p*-values < 0.001), indicating that higher self-reported PM Abilities tended to be associated with more frequent use of both internal and external PM strategies. A highly significant, positive moderate correlation was also found between scores on the Internal PM strategies component of the MPMI-s and scores on the External PM strategies component, *p* < 0.001. This indicates that participants who reported the use of more internal PM strategies also tended to report the use of more external PM strategies.

### 3.2. Hierarchical Multiple Regression Analysis

No multicollinearity concerns were identified, as the correlations among the predictor variables were below the commonly accepted threshold of *r* = 0.80 (e.g., [76,77]). Therefore, separate hierarchical multiple regression analyses were subsequently performed.

#### 3.2.1. Executive Functioning

The analysis conducted on the overall score on the BDEFS is summarised in Table 4.

For the overall BDEFS scores, regression revealed that in Block 1, age contributed significantly to the regression model, *F*(1, 166) = 6.59, *p* = 0.011, accounting for 3.8% of the variance in scores. An additional 51.7% of the variance in scores was explained upon the introduction of the ASRS scores in Block 2 and this *R*^2^ change was highly significant, *F*(2, 165) = 102.85, *p* < 0.001. The introduction of the ARQ scores in Block 3 accounted for an additional 4% of the variance in scores, and the predictive power of the model remained highly significant, *F*(3, 164) = 80.12, *p* < 0.001. Age, ASRS scores and ARQ scores were significant, positive predictors of overall BDEFS scores. The latter relationship is shown in Figure 1.

The analyses conducted on the cool subcomponents of the BDEFS are summarised in Table 5.

In the case of the Self-Management to Time component of the BDEFS, age was not a significant predictor, accounting for only 1.6% of the variance in scores. The predictive power of the model in Block 1 was not statistically significant, *F*(1, 166) = 2.63, *p* = 0.107. An additional 55.7% of the variance in Self-Management to Time scores was explained after the introduction of the ASRS into the model, and this contribution was significant, *F*(2, 165) = 110.73, *p* < 0.001. The addition of the ARQ to the model accounted for an additional 2.9% of the variance in scores, and this *R*^2^ change was highly significant, *F*(3, 164) = 82.71, *p* < 0.001. Figure 2 illustrates the positive association between ARQ scores and Self-Management to Time scores.

Considering the Self-Organisation/Problem Solving component of the BDEFS, age made a significant contribution to the regression model, *F*(1, 166) = 5.54, *p* = 0.020, accounting for 3.2% of the variance in scores. An additional 44.1% of the variance in scores was explained following the introduction of the ASRS and this *R*^2^ change was highly significant *F*(2, 165) = 74.00, *p* < 0.001. In Block 3, the predictive power of the model remained highly significant, *F*(3, 164) = 69.05, *p* < 0.001. ARQ scores explained an additional 8.5% of the variance in Self-Organisation/Problem Solving scores. Age was a significant, negative predictor of Self-Organisation/Problem Solving scores, whereas ASRS and ARQ scores were significant, positive predictors. Figure 3 shows the positive relationship between ARQ scores and scores on the Self-Organisation/Problem Solving component of the BDEFS.

For the Self-Restraint/Inhibition component of the BDEFS, the predictive power of the model in Block 1 was not significant, *F*(1, 166) = 3.32, *p* = 0.070. Age accounted for only 2% of the variance in scores. However, the introduction of the ASRS in Block 2 accounted for an additional 32.5% of the variance in scores and the *R*^2^ change was found to be significant *F*(2, 165) = 43.29, *p* < 0.001. In Block 3, ARQ scores contributed significantly to the regression model, *F*(3, 164) =31.13, *p* < 0.001, accounting for 1.9% of the variance in Self-Restraint/Inhibition scores. Figure 4 illustrates the positive association between ARQ scores and scores on the Self-Restraint/Inhibition component of the BDEFS.

The analyses conducted on the hot subcomponents of the BDEFS are summarised in Table 6.

In the case of the Self-Motivation component of the BDEFS, age was a significant, negative predictor, *F*(1, 166) = 8.12, *p* = 0.005, accounting for 4.7% of the variance in scores on that component. The addition of the ASRS to the model explained a further 34.5% of the variance in Self-Motivation scores and the *R*^2^ change was found to be significant *F*(2, 165) = 53.01, *p* < 0.001. Scores on the ASRS were a significant, positive predictor of Self-Motivation scores. The introduction of the ARQ in Block 3 accounted for an increase of only 0.9% in the variance in scores and this *R*^2^ change was not statistically significant. Although these results indicate that age and ASRS scores significantly predicted Self-Motivation scores, whereas ARQ scores did not, the overall predictive power of the model in Block 3 remained highly significant, *F*(3, 164) = 36.54, *p* < 0.001.

For the Self-Regulation of Emotion component of the BDEFS, age contributed significantly to the regression model, *F*(1, 166) = 7.11, *p* = 0.008, accounting for 4.1% of the variance in Self-Regulation of Emotion scores. The introduction of the ASRS scores in Block 2 accounted for an additional 16.6% of the variance in scores, and this *R*^2^ change was highly significant, *F*(2, 165) = 21.50, *p* < 0.001. In Block 3, the addition of the ARQ scores accounted for an additional 0.6% of the variance in scores and this *R*^2^ change was not statistically significant. However, the overall predictive power of the model remained highly significant, *F*(3, 164) = 14.74, *p* < 0.001. Age was a significant, negative predictor of Self-Regulation of Emotion scores, whereas ASRS scores were significant, positive predictors.

#### 3.2.2. Prospective Memory

The test statistics for each of the hierarchical regressions performed on the components of the MPMI-s are presented in Table 7.

There was a negative relationship between the two predictor variables (ASRS and ARQ) and all MPMI-s components. Higher scores on the ASRS and ARQ predicted lower scores on the PM Abilities, Internal PM strategies and External PM strategies components of the MPMI-s. Age was negatively associated with PM Abilities, but positively associated with both strategy components of the MPMI-s.

For the PM Abilities component of the MPMI-s, age in Block 1 accounted for 0.2% of the variance in scores and this contribution to the model was not significant, *F*(1, 166) = 0.41, *p* = 0.523. An additional 33% of the variance in scores was explained upon the introduction of the ASRS into the model. The *R*^2^ change was statistically significant, *F*(2, 165) = 41.01, *p* < 0.001. In Block 3, the predictive power of the model remained highly significant, *F*(3, 164) = 32.89, *p* < 0.001. ARQ scores explained an additional 4.4% of the variance in PM Abilities component scores. Figure 5 illustrates the negative association between ARQ scores and PM Abilities scores, with greater self-reported dyslexia traits tending to be related to lower self-perceived PM Abilities.

For the Internal PM Strategies component of the MPMI-s, age contributed to 0.8% of the variance in scores, but this *R*^2^ change was not significant, *F*(1, 166) = 1.36, *p* = 0.246. Scores on the ASRS did not contribute to the variance in scores, thus the predictive power of the model was not significant, *F*(2, 165) = 0.693, *p* = 0.501. Similarly, the introduction of the ARQ in Block 3 did not account for any additional variance in scores, thus the predictive power of the model was also not significant *F*(3, 164) = 0.470, *p* = 0.704.

On the External PM Strategies component of the MPMI-s, age was found to be a significant, positive predictor, accounting for 5% of the variance in scores. The overall predictive power of the model in Block 1 was significant, *F*(1, 166) = 8.71, *p* = 0.004. Scores on the ASRS accounted for just 0.8% of the variance in scores and although this *R*^2^ change was not significant, the overall predictive power of the model remained significant, *F*(2, 165) = 5.06, *p* = 0.007. The introduction of the ARQ did not account for any additional variance in scores and was, thus, it was not a significant predictor of External PM Strategies scores. However, the predictive power of the model in Block 3 remained significant overall, *F*(3, 164) = 3.35, *p* = 0.020.

## 4. Discussion

The present study aimed to use self-report measures to contribute to the limited literature focusing on dyslexia traits in relation to cognition in adulthood, particularly the influence dyslexia traits may have on hot and cold EFs, PM and the use of PM offloading strategies. Overall, after controlling for age and potential co-occurring ADHD symptoms, participants who self-reported higher levels of dyslexia traits also tended to report more frequent EF problems and tended to give lower ratings of their PM Abilities. These findings were consistent with the overarching hypotheses. However, higher levels of dyslexia traits were not a significant predictor of either internal or external PM strategy use, which was not consistent with the proposed hypotheses.

The results demonstrated a significant relationship between EF deficits and dyslexia-related traits. This finding is in line with previous research that has identified EF deficits in adults with officially diagnosed dyslexia (e.g., [8,12]). When looking at the individual EFs measured, participants reporting higher levels of dyslexia traits tended to report more frequent everyday problems in time-management and organisation/problem-solving, supporting the hypothesis that dyslexia traits would be associated with deficits in cool EFs. Such skills can be measured in isolation in the laboratory, but rigid, well-defined laboratory tasks do not represent the way in which higher-order cognition is typically called upon in real-world environments (e.g., [78,79]). Nonetheless, the findings are consistent with the reports of dyslexia practitioners identifying persistent poor performance in adults with dyslexia in time management and estimation, and in organisation skills (e.g., [80,81,82]). From a theoretical perspective, the findings challenge the dominant dyslexia theories (e.g., [3,4]) that do not explain EF difficulties within their frameworks. Higher self-reported dyslexia traits were also associated with more frequently reported problems relating to self-restraint/inhibition. This finding does not support the overarching hypothesis that dyslexia influences cool but not hot EFs, as in the current study, inhibition was classified as hot EF due to the emotional and motivational significance inherent in the relevant questions within the BDEFS. Although there is some evidence in the literature to suggest that children and adults with dyslexia have difficulty with inhibitory control and response inhibition when task demands align with cool EF characteristics (e.g., [8,83]), the findings of the present study suggest that challenges may also extend to more motivationally or emotionally charged contexts. Further empirical investigation is warranted to clarify whether inhibition difficulties in dyslexia are task- or context-dependent, to develop a more comprehensive understanding of the cognitive and emotional regulation processes underlying inhibition in dyslexia.

The remaining EFs measured did not yield any significant associations with dyslexia traits. Although there was a significant positive bivariate relationship between increased self-reported dyslexia traits and more frequent self-reported problems with Self-Motivation and Self-Regulation of Emotion, dyslexia traits were not a significant predictor of the two aforesaid EFs. These findings are consistent with Smith-Spark et al. [12] who found no group differences on the Behavioural Regulation Index of the BRIEF-A [28] in adults with a confirmed diagnosis of dyslexia. Motivation and regulation of emotion are hot EFs, and the literature suggests that these may be impaired in ADHD rather than dyslexia (e.g., [35,61,62,63]). Indeed, higher self-reported ADHD symptomatology significantly predicted more frequently reported problems with self-motivation and regulation of emotion, further emphasising the notion that both cool and hot EFs are likely to be impaired in ADHD. Whether this is the case in dyslexia is still unclear and warrants further empirical investigation, using laboratory-based tasks that measure hot EFs and their underlying processes, such as reward-based tasks or (risky) monetary decision-making tasks [23].

The current study has also identified a significant relationship between dyslexia traits and PM Abilities, with participants who reported higher levels of dyslexia traits also tending to give lower ratings of their PM Abilities. This finding supports previous work that has identified PM deficits in dyslexia, using self-reports [9,13], under laboratory conditions [14], using a combination of self-report and laboratory measures [15], through interviews [56] and even in research incorporating naturalistic tasks of PM (e.g., [14,15]). While self-report measures have been criticised for lacking validity when indicating PM ability [84] and, in a review, Sugden et al. [85] highlight no significant relationships between self-report and performance-based measures of PM, the current study, in combination with the other published literature on the role of dyslexia on PM, seems to contradict those views, since the findings from self-reports and laboratory tasks are in general agreement in this case (albeit at different levels of cognition; see [43,86]).

When considering cognitive offloading, dyslexia traits did not significantly predict the use of either internal or external PM strategies. A negative trend was observed between dyslexia traits and strategy use; participants who reported more dyslexia-related traits tended to report less frequent use of internal and external PM strategies, although this relationship was very weak and non-significant. The absence of a significant finding was not anticipated, as the expectation was that participants who self-reported more PM difficulties would also self-report the use of more strategies to help them with those difficulties. However, significant positive relationships were found between PM Abilities and strategy use, indicating that respondents who gave higher ratings of their PM Abilities, also tended to report the use of more internal and external PM strategies. These findings are broadly consistent with those of Bacon et al. [54], who documented the challenges faced by adults with diagnosed dyslexia in voluntarily initiating effective strategies during task performance. Such difficulties may lead to a poorer and less effective use of available aids.

Although the findings of the current study are not in line with Smith-Spark et al. [15], who found that adults with diagnosed dyslexia self-reported the use of more techniques to assist memory, the participants with greater dyslexia traits are, arguably, not using those techniques successfully, since they still reported more frequent PM difficulties (as argued previously by Smith-Spark et al. [15] and Smith-Spark [87]). While it is not possible to determine this conclusively based on the current data, individuals with dyslexia may be aware that a new technique is required but the difficulty may lie in generating an alternative schema or adapting an existing one in order to inhibit an already established, yet ineffective, technique [88]. Indeed, Meltzer [89] suggested that individuals with dyslexia potentially lack the ability to access metacognitive information efficiently, which could in turn hinder their ability to even recognise the effectiveness of particular useful strategies. Therefore, it is plausible that individuals with dyslexia, although self-reporting more PM difficulties in everyday life, may lack the metacognitive ability to initiate or appropriately apply strategies that would benefit their prospective remembering. On the basis of these findings, and assuming that the same pattern is observed in adults with a formal diagnosis (e.g., [15]), there is a clear need for this to be explored in more depth experimentally and also to develop relevant interventions. Strategy-related training could be developed for individuals with dyslexia to inform them of available strategies that could help with their everyday memory failures and to coach them on the most efficient way to utilise them, in order to fully benefit from them. Although the literature is limited, research on the effectiveness of coaching in dyslexia has indicated that such interventions can be beneficial (e.g., [90]), providing scope for more targeted coaching interventions to be developed and implemented, to improve outcomes for adults with dyslexia.

Although the findings of the study highlight important implications for dyslexia theory and practice, there are several limitations to note. While efforts were made to recruit a wide sample, there was a substantial gender imbalance in the final sample (33 male, 145 female). There is a growing literature reporting gender differences in executive functioning (see [91] for a review), although these effects are often task-dependent and may reflect outcome preference or strategy rather than underlying ability (e.g., [92]). As a consequence, although caution should be exercised when generalising the results, it is unlikely that the gender imbalance is the primary driver of the observed effects. Similarly, only ADHD symptomatology was statistically controlled for in the analyses and, while participants who reported other diagnoses were excluded from the sample, there still remains a possibility that other factors, such as negative emotional consequences [93], could have contributed to the observed deficits, in addition to dyslexia traits. Moreover, there is a potential for response bias and demand characteristics, due to participants being aware of the study’s focus on dyslexia. To minimise this risk, explicit references to dyslexia in study advertisements were replaced with more neutral terms, such as “reading and spelling abilities”, although dyslexia was directly mentioned in the participant-facing documents, so the possibility of some bias remains. Only subjective, self-report measures were used in the study, so arguably the data may lack validity as the approach is reliant on the participants providing candid self-reports. However, the Cronbach’s alpha values and the results of the present study were comparable and broadly consistent with the previous literature, utilising a range of different methodologies to fully understand the effects of dyslexia “in the wild”, via a process of triangulation. Nonetheless, the data were carefully examined, and incomplete or seemingly unreliable responses were excluded from the analysis. The use of self-report measures was adopted in order to utilise more ecologically valid measures of EF and PM, which capture additional environmental and psychological aspects of these multidimensional constructs, that may not otherwise be measured by experimental tasks in the laboratory [4,94]. Moreover, self-report measures of dyslexia traits provide a means of accessing community samples and older individuals, who may have never had a formal diagnosis of dyslexia yet still experience symptoms in daily life. It is important to give a voice to such individuals (who may be more difficult to access than younger, student participants due to likely greater work and family commitments) and to gain insights into their everyday cognition.

The present study has enhanced the current understanding of the role of dyslexia traits on the perceived effectiveness of everyday EF and PM, by exploring multiple domains of EF as well as the use of PM strategies. Future research could focus on further exploring the relationship between dyslexia traits or diagnosis, and time-management, organisation and inhibition, by utilising a range of methodologies to reach more accurate conclusions. Since the BDEFS has yielded some significant results, it might be beneficial to be used in the future as a tool in dyslexia assessment (although still controlling for co-occurring ADHD), perhaps not in a diagnostic manner, but as a complementary tool that would allow clinicians to better understand the cognitive impact of dyslexia. The BDEFS has been used in relation to realistic career choices and planning in individuals with ADHD [95] and it could be used in a similar way with individuals with dyslexia. By identifying their strengths and difficulties using the BDEFS, individuals with dyslexia can be empowered to select a career path which celebrates their strengths and may also allow them to work on improving some of their persistent difficulties. Future research on PM and dyslexia could focus on more ecologically valid paradigms to measure PM in everyday life. The emerging literature utilises diary and ecological momentary assessment measures (e.g., [96,97]) to capture PM in more natural environments. At present, such paradigms have not been used in relation to dyslexia, with the exception of Smith-Spark’s [98] diary study with formally diagnosed students, thereby providing scope for more naturalistic contributions to the literature.

## 5. Conclusions

The present study supports the argument that dyslexia symptoms (regardless of whether or not dyslexia has been formally diagnosed) may be associated with broader cognitive deficits that extend beyond the literacy domain, as it has identified more frequent self-reported difficulties with time-management, organisation, self-restraint and PM in individuals with elevated dyslexia traits. Assuming that this pattern extends to individuals with an official diagnosis of dyslexia, these difficulties may make it challenging for adults with dyslexia to follow through on intentions and manage everyday tasks. Increased everyday EF and PM errors may therefore serve as a marker for identifying individuals who could benefit from diagnostic testing, particularly adults who do not engage in significant amounts of reading or writing but nonetheless may need support with the broader cognitive difficulties associated with dyslexia. Understanding and addressing these cognitive challenges is crucial for supporting adults with elevated dyslexia traits to achieve their full potential.

## Figures and Tables

**Figure 1 brainsci-15-01065-f001:**
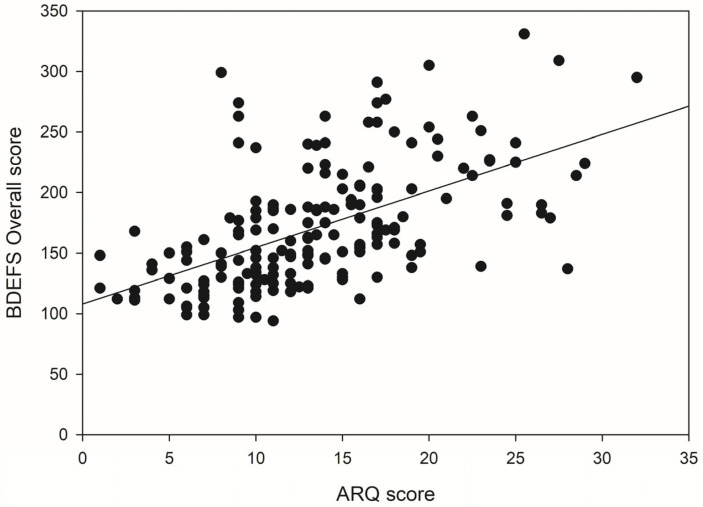
The relationship between ARQ scores and overall BDEFS scores.

**Figure 2 brainsci-15-01065-f002:**
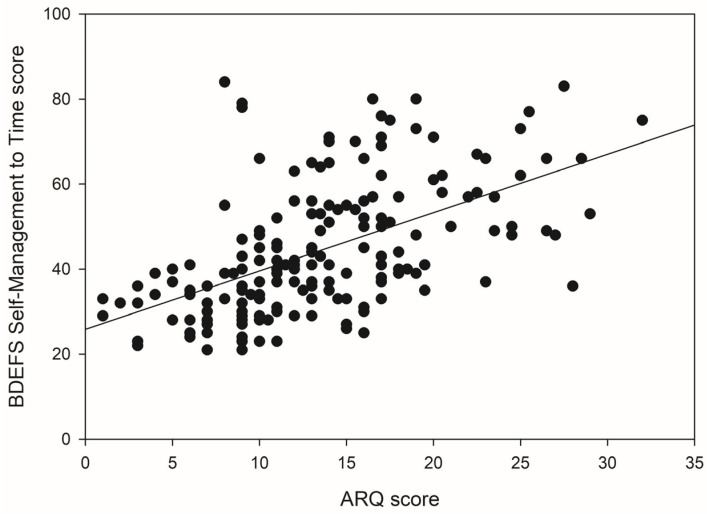
The relationship between ARQ scores and scores on the BDEFS Self-Management to Time subcomponent.

**Figure 3 brainsci-15-01065-f003:**
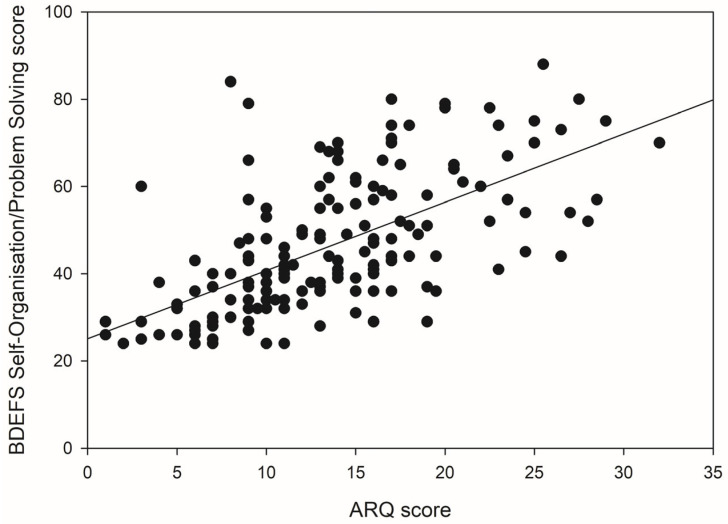
The relationship between ARQ scores and scores on the BDEFS Self-Organisation/Problem Solving subcomponent.

**Figure 4 brainsci-15-01065-f004:**
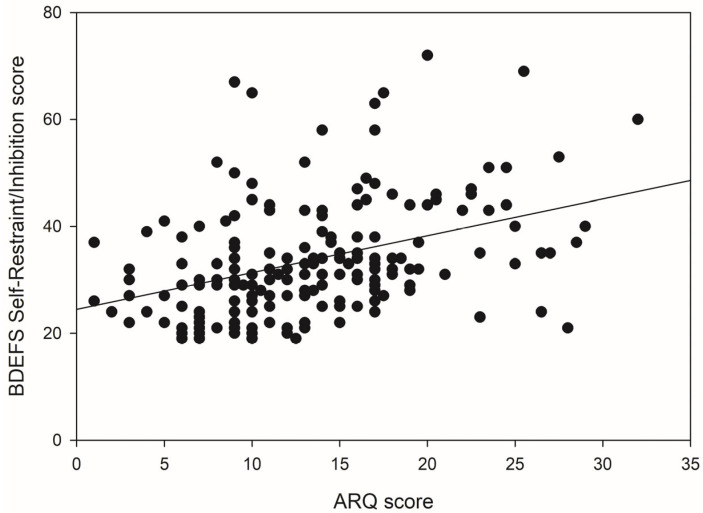
The relationship between ARQ scores and scores on the BDEFS Self-Restraint/Inhibition component.

**Figure 5 brainsci-15-01065-f005:**
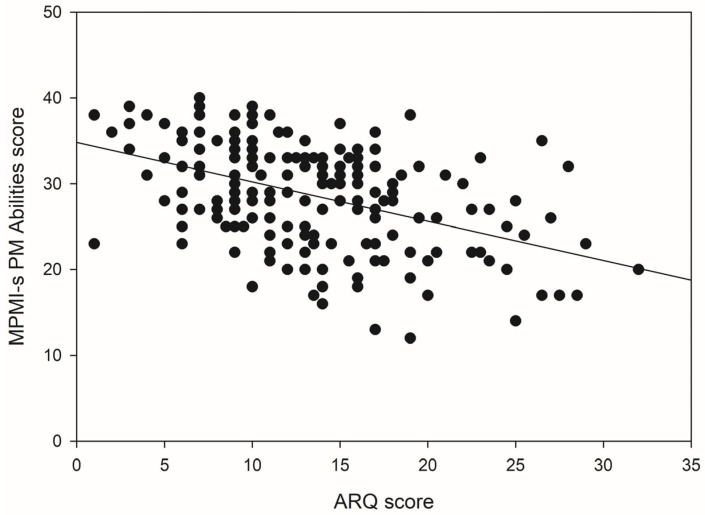
The relationship between ARQ scores and scores on the MPMI-s PM Abilities component.

**Table 1 brainsci-15-01065-t001:** Frequency counts for the background measures obtained.

Background Measure	Number of Participants
Students (not in employment)	48
Students (in employment)	84
Employed (not currently a student)	45
Unemployed/Retired	3
Completed secondary school/college	75
Completed undergraduate degree	69
Completed postgraduate degree (Masters or Doctoral level)	36

**Table 2 brainsci-15-01065-t002:** Descriptive statistics and reliability coefficients for the self-report questionnaires.

Questionnaire Measure	Minimum	Maximum	Mean	*SD*	Cronbach’s *α*
ARQ	1	32	13.44	6.10	0.76
ASRS	0	6	2.66	1.95	0.74
BDEFS—Overall	94	331	170.73	50.52	0.98
BDEFS—Self-Management to Time	21	84	44.30	15.38	0.97
BDEFS—Self-Organisation/Problem Solving	24	88	46.13	15.33	0.95
BDEFS—Self-Restraint/Inhibition	19	72	33.72	10.87	0.93
BDEFS—Self-Motivation	12	48	20.68	8.14	0.94
BDEFS—Self-Regulation of Emotion	13	52	25.90	9.83	0.95
MPMI-s—PM Abilities	12	40	28.64	6.09	0.79
MPMI-s—Internal PM Strategies	7	35	23.18	6.21	0.84
MPMI-s—External PM Strategies	7	35	24.29	6.65	0.84

**Table 3 brainsci-15-01065-t003:** Correlation matrix of all variables.

	1.	2.	3.	4.	5.	6.	7.	8.	9.	10.	11.	12.
**1. Age**	-											
**2. ARQ**	−0.23 **	-										
**3. ASRS**	−0.10	0.52 ***	-									
**4. BDEFS Overall**	−0.20 *	0.56 ***	0.73 ***	-								
**5. BDEFS Self-Management to Time**	−0.13	0.54 ***	0.75 ***	0.89 ***	-							
**6. BDEFS Self-Organisation/Problem Solving**	−0.18 *	0.62 ***	0.68 ***	0.87 ***	0.73 ***	-						
**7. BDEFS Self-Restraint/Inhibition**	−0.14	0.39 ***	0.56 ***	0.83 ***	0.63 ***	0.57 ***	-					
**8. BDEFS Self-Motivation**	−0.22 **	0.43 ***	0.61 ***	0.85 ***	0.76 ***	0.69 ***	0.66 ***	-				
**9. BDEFS Self-Regulation of Emotion**	−0.20 **	0.29 ***	0.41 ***	0.78 ***	0.52 ****	0.57 ***	0.74 ***	0.56 ***	-			
**10. MPMI-s PM Abilities**	−0.05	−0.46 ***	−0.57 ***	−0.59 ***	−0.69 ***	−0.46 ***	−0.50 ***	−0.49 ***	−0.32 ***	-		
**11. MPMI-s Internal PM Strategies**	0.09	−0.03	−0.02	−0.01	−0.08	0.08	−0.07	−0.03	0.04	0.36 ***	-	
**12. MPMI-s External PM Strategies**	0.22 **	−0.09	−0.12	−0.04	−0.15 *	0.07	−0.05	−0.07	0.03	0.30 ***	0.53 ***	-

Key: * = *p* ≤ 0.05; ** = *p* ≤ 0.01; *** = *p* ≤ 0.001.

**Table 4 brainsci-15-01065-t004:** Summary of the hierarchical multiple regression analysis for variables predicting overall BDEFS scores.

	Variable	β (Standardised)	*t*	*R*	*R*^2^ (Adjusted)	Δ*R*^2^
**BDEFS Overall**	Block 1			0.195	0.032	0.038
	Age	−0.195	−2.568 *			
	Block 2			0.745	0.550	0.517
	Age	−0.126	−2.407 *			
	ASRS	0.722	13.840 ***			
	Block 3			0.771	0.587	0.040
	Age	−0.082	−1.610 ^#^			
	ASRS	0.603	10.367 ***			
	ARQ	0.238	3.997 ***			

Key: * = *p* ≤ 0.05; *** = *p* ≤ 0.001; ^#^ = overall regression model was not significant.

**Table 5 brainsci-15-01065-t005:** Summary of the hierarchical multiple regression analyses for variables predicting cool BDEFS subcomponent scores.

	Variable	β (Standardised)	*t*	*R*	*R*^2^ (Adjusted)	Δ*R*^2^
**BDEFS** **Self-Management to Time**	Block 1			0.125	0.010	0.016
	Age	−0.125	−1.623 ^#^			
	Block 2			0.757	0.568	0.557
	Age	−0.052	−1.026 ^#^			
	ASRS	0.750	14.677 ***			
	Block 3			0.776	0.595	0.029
	Age	−0.015	−0.302 ^#^			
	ASRS	0.648	11.246 ***			
	ARQ	0.204	3.459 ***			
**BDEFS** **Self-Organisation/** **Problem Solving**	Block 1			0.180	0.026	0.032
	Age	−0.180	−2.353 *			
	Block 2			0.688	0.466	0.441
	Age	−0.115	−2.028 *			
	ASRS	0.667	11.743 ***			
	Block 3			0.747	0.550	0.085
	Age	−0.052	−0.966 ^#^			
	ASRS	0.492	8.101 ***			
	ARQ	0.349	5.626 ***			

Key: * = *p* ≤ 0.05; *** = *p* ≤ 0.001; ^#^ = overall regression model was not significant.

**Table 6 brainsci-15-01065-t006:** Summary of the hierarchical multiple regression analyses for variables predicting hot BDEFS subcomponent scores.

	Variable	β (Standardised)	*t*	*R*	*R*^2^ (Adjusted)	Δ*R*^2^
**BDEFS** **Self-Restraint/Inhibition**	Block 1			0.140	0.014	0.020
	Age	−0.140	−1.822 ^#^			
	Block 2			0.587	0.336	0.325
	Age	−0.085	−1.337 ^#^			
	ASRS	0.572	9.036 ***			
	Block 3			0.602	0.351	0.019
	Age	−0.055	−0.856 ^#^			
	ASRS	0.490	6.726 ***			
	ARQ	0.164	2.194 *			
**BDEFS Self-Motivation**	Block 1			0.216	0.041	0.047
	Age	−0.216	−2.849 **			
	Block 2			0.625	0.384	0.345
	Age	−0.159	−2.604 **			
	ASRS	0.590	9.664 ***			
	Block 3			0.633	0.390	0.009
	Age	−0.138	−2.216 *			
	ASRS	0.532	7.516 ***			
	ARQ	0.116	1.607 ^#^			
**BDEFS** **Self-Regulation of Emotion**	Block 1			0.203	0.035	0.041
	Age	−0.203	−2.667 **			
	Block 2			0.455	0.197	0.166
	Age	−0.163	−2.342 *			
	ASRS	0.409	5.870 ***			
	Block 3			0.461	0.198	0.006
	Age	−0.147	−2.060 *			
	ASRS	0.364	4.489 ***			
	ARQ	0.090	1.084 ^#^			

Key: * = *p* ≤ 0.05; ** = *p* ≤ 0.01; *** = *p* ≤ 0.001; ^#^ = overall regression model was not significant.

**Table 7 brainsci-15-01065-t007:** Summary of the hierarchical multiple regression analyses for variables entered as potential predictors of MPMI-s scores.

	Variable	β (Standardised)	*t*	*R*	*R*^2^ (Adjusted)	Δ*R*^2^
**MPMI-s** **PM Abilities**	Block 1			0.050	−0.004	0.002
	Age	−0.050	−0.641 ^#^			
	Block 2			0.576	0.324	0.330
	Age	−0.105	−1.649 ^#^			
	ASRS	−0.577	−9.023 ***			
	Block 3			0.613	0.364	0.044
	Age	−0.151	−2.379 *			
	ASRS	−0.452	−6.258 ***			
	ARQ	−0.250	−3.384 ***			
**MPMI-s** **Internal PM Strategies**	Block 1			0.090	0.002	0.008
	Age	0.090	1.164 ^#^			
	Block 2			0.091	−0.004	0.000
	Age	0.089	1.136 ^#^			
	ASRS	−0.015	−0.198 ^#^	0.092	−0.010	0.000
	Block 3					
	Age	0.086	1.070 ^#^			
	ASRS	−0.007	−0.080 ^#^			
	ARQ	−0.016	−0.175 ^#^			
**MPMI-s** **External PM Strategies**	Block 1			0.223	0.044	0.050
	Age	0.223	2.951 **			
	Block 2			0.240	0.046	0.008
	Age	0.215	2.826 **			
	ASRS	−0.089	−1.179 ^#^			
	Block 3			0.240	0.041	0.000
	Age	0.214	2.748 **			
	ASRS	−0.088	−0.996 ^#^			
	ARQ	−0.002	−0.026 ^#^			

Key: * = *p* ≤ 0.05; ** = *p* ≤ 0.01; *** = *p* ≤ 0.001; ^#^ = overall regression model was not significant.

## Data Availability

The original data presented in the study are openly available in the Open Science Framework at https://osf.io/5ety4/ (accessed date 21 September 2025).

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
