# Peer review of "Adult Dyslexia Traits as Predictors of Hot/Cool Executive Function and Prospective Memory Abilities"

_brainsci, 2025, doi:10.3390/brainsci15101065_

Round 1
Reviewer 1 Report
Comments and Suggestions for Authors
Please finde my review in the attachment.

Reviewer 2 Report
Comments and Suggestions for Authors
This article examines the relationship between adult dyslexia and executive functioning (EF), prospective memory (PM), and compensatory strategies (use of internal or external aids). The authors also consider symptoms of Attention Deficit Hyperactivity Disorder (ADHD), which frequently co-occur with dyslexia.
The findings suggest that adult dyslexia extends beyond reading and writing difficulties: it also affects time management, organization, inhibition, and prospective memory. Deficits appear more pronounced for cool EF (cognitive control) than for hot EF (emotion- and motivation-related control). Nevertheless, the absence of an increased use of compensatory strategies, despite clear difficulties, points toward a possible metacognitive deficit. The authors conclude by proposing several avenues for practical interventions.
Methodologically, the study is well designed: the sample size is relatively large (N = 180), the tools used are appropriate (ARQ, ASRS, BDEFS, MPMI-s), and these psychometric instruments have been scientifically validated with good internal consistency. The statistical analyses (hierarchical regressions) are sound and carefully conducted, with age and ADHD controlled for.
The article is particularly valuable because it promotes a more ecological approach to adult dyslexia. Adults with dyslexia are an understudied population, and the focus here is on everyday experiences rather than solely on laboratory-based tests. In addition, the paper demonstrates practical relevance by engaging with issues of support, coaching, and the potential use of complementary assessment tools.
Critical points :
Theoretical concerns
The hot/cool EF distinction is somewhat questionable. For instance, inhibition is classified as “hot,” even though it encompasses both cool (cognitive control) and hot (affective) aspects (see manuscript, line 108). The authors could strengthen their argument by addressing this apparent contradiction.
Sample limitations
The sample is gender-imbalanced (145 women / 33 men). Since gender differences in executive functioning have been documented, this limits the generalizability of the results. It would be useful for the authors to acknowledge this issue explicitly. For reference:
-
-
- Grissom, N.M., & Reyes, T.M. (2019). Let’s call the whole thing off: evaluating gender and sex differences in executive function. Neuropsychopharmacology, 44, 1–11. https://doi.org/10.1038/s41386-018-0179-5
- Gaillard, A., Fehring, D. J., & Rossell, S. L. (2021). Sex differences in executive control: A systematic review of functional neuroimaging studies. European Journal of Neuroscience, 53(8), 2592–2611. https://doi.org/10.1111/ejn.15107
-
Only 12 participants had an official diagnosis of dyslexia. While it is understandable that adult dyslexia is difficult to assess (due to the lack of standardized diagnostic tools), the authors should exercise greater caution in using the term dyslexia. Their reliance on the phrase dyslexia traits is appropriate, but in a few places (e.g., conclusion, line 681), the wording implies confirmed diagnosis.
Other considerations
Only ADHD was controlled for, whereas dyslexia often co-occurs with other conditions such as dyspraxia, dyscalculia, motor coordination difficulties, and mental health issues (e.g., anxiety, depression).
Although the authors avoid the common pitfall of equating correlation with causation, it remains important to stress that dyslexia traits cannot be assumed to cause the observed deficits.
Conclusion
Overall, this is an interesting and well-constructed study that extends the scope of research on adult dyslexia beyond literacy, highlighting difficulties with time management, organization, inhibition, and prospective memory.
However, the conclusions should remain cautious, as they are based on subjective, correlational data from a somewhat biased sample. Despite these limitations, the study opens up promising avenues for further research and for the development of practical interventions.
Round 2
Reviewer 1 Report
Comments and Suggestions for Authors
Dear authors,
Thank you for accommodating most of my comments and pardon for my misunderstanding about the age.
One could perform a multiple regression to analyse more than one DVs in regression. Therefore, I still recommend to use this analysis. However, if you provide argument(s) that simple linear regression is better than multiple regression in this study (I am sorry I cannot help with the argument), the readers may be convinced.
Round 3
Reviewer 1 Report
Comments and Suggestions for Authors
Thank you for your explanation. I may have distinct opinion about how to use multilevel regression but I think you have also your opinion. I think if you explain it in the manuscript, it would be sufficient.